# Transcriptome Analysis Unveiled the Intricate Interplay between Sugar Metabolism and Lipid Biosynthesis in *Symplocos paniculate* Fruit

**DOI:** 10.3390/plants12142703

**Published:** 2023-07-20

**Authors:** Wenjun Li, Lijuan Jiang, Yunzhu Chen, Changzhu Li, Peiwang Li, Yan Yang, Jingzhen Chen, Qiang Liu

**Affiliations:** 1College of Life Science and Technology, Central South University of Forestry and Technology, Changsha 410004, China; lwjlwjlwj1998@163.com (W.L.); znljiang2542@163.com (L.J.); 2State Key Laboratory of Utilization of Woody Oil Resource, Hunan Academy of Forestry, Changsha 410004, China; cyzcarol@foxmail.com (Y.C.); lichangzhu2013@aliyun.com (C.L.); lindan523@163.com (P.L.); yangyanzupei@126.com (Y.Y.)

**Keywords:** *Symplocos paniculate*, sugar metabolism, lipid biosynthesis, transcriptome, differential expression

## Abstract

*Symplocos paniculate* is an oil plant exhibiting tissue-specific variations in oil content and fatty acid composition across the whole fruit (mainly pulp and seed). And its oil synthesis is intricately linked to the accumulation and transformation of sugars. Nevertheless, there remains a dearth of understanding regarding how sugar metabolism impacts oil synthesis in *S. paniculate* fruit. To unravel the intricate mechanism underlying the impact of sugar metabolism on lipid biosynthesis in *S. paniculata* fruit, a comparative analysis was conducted on the transcriptome and metabolite content of pulp and seed throughout fruit development. The findings revealed that the impact of sugar metabolism on oil synthesis varied across different stages of fruit development. Notably, during the early fruit developmental stage (from 90 to 120 DAF), pivotal genes involved in sugar metabolism, such as PGK3, PKP1, PDH-E1, MDH, and malQ, along with key genes associated with oil synthesis like KAR, HAD, and PAP were predominantly expressed in the pulp. Consequently, this preferential expression led to earlier accumulation of oil in the pulp tissue compared to the seed. Whereas, during the fruit maturity stage (from 120 DAF to 140 DAF), these genes exhibited a high level of expression in seed, thereby facilitating the rapid and substantial accumulation of seed oil compared to pulp. The sugar metabolism activity in various parts of *S. paniculata* fruit plays a pivotal role in oil synthesis and is contingent upon the developmental stage. These findings can offer alternative genes for further gene enhancement through molecular biotechnology, thereby augmenting fruit oil yield and altering fatty acid composition.

## 1. Introduction

*Symplocos paniculate*, a member of the *Symplocaceae* family, is a deciduous shrub or small tree indigenous to eastern Asia, encompassing China, Japan, and Korea. It boasts ecological and economic significance as its entire fruit contains 36.6% oil with an unsaturated fatty acid content of 79.8% [1]. As such, *S. paniculata* serves as an optimal feedstock for both bio-diesel and edible oil production [2]. Furthermore, its ability to withstand severe drought, high salinity, and alkalinity in soils makes it a crucial contributor to maintaining ecosystem function by preventing desertification and erosion. This quality also renders it ideal for cultivation on marginal land [3]. Oil is the primary metabolite of *S. paniculate* fruit, and its oil content and quality are pivotal determinants of economic value. Therefore, comprehending the biosynthesis pathway and regulatory mechanism of oil is paramount to enhancing both the quantity and composition of fruit in oil plants.

Numerous studies have demonstrated that oil biosynthesis is not an independent process, but rather a complex interplay of carbon flux distribution involving glycometabolism, fatty acid biosynthesis, and triacylglycerols (TAG) assembly [4]. As we are aware, carbohydrate metabolism plays a pivotal role in fruit development by providing cellulose and hemicellulose for cell structure formation and energy for fruit growth as well as the carbon framework of intermediate products (pyruvyl) [5].

In general, the metabolism of carbohydrates is intimately linked to the biosynthesis of lipids. During the development of rapeseed seeds, a gradual conversion of soluble sugars into fatty acids and oils for storage was observed [6]. This period of change in the soluble sugar component during oil seed development often impacts on later seed oil accumulation [7]. And in the process of oil accumulation, the intermediate products of glucose metabolism (glycolysis or pentose phosphate pathway) serve as the primary carbon source for fatty acid synthesis and oil accumulation [8]. Furthermore, it was discovered that Glucose-6-phosphate and Triose-phosphate constitute the principal carbon sources for fatty acid synthesis [9]. The enhancement of seed oil accumulation can be achieved through the alteration of gene expression associated with sugar metabolism [10]. However, the relationship between oil and sugar may vary among different plant species due to their distinct genetic inheritance [11]. For instance, during the fruit development of *Xanthoceras sorbifolia*, changes in oil content were negatively correlated with sugar content, suggesting that oil was primarily derived from carbohydrate conversion [12]. Whereas study of relevance has discovered that there is no significant correlation between oil content and sugar levels in rapeseed’s sugar content [13]. Therefore, analyzing the effect of sugars on lipid synthesis solely through correlation analysis of sugar and oil content changes is inadequate due to their direct or indirect effects on oil synthesis.

Furthermore, the carbohydrate metabolic pathways encompassing glycolysis, starch anabolism, gluconeogenesis, and the tricarboxylic acid cycle exhibit partial independence from one another, and the existence of absolute substrate competition remains a topic of controversy. Theoretically, the synthesis of starch would decrease the availability of monosaccharides necessary for glycolysis, thereby creating a competition with oil synthesis [14]. However, numerous studies have demonstrated that starch storage does not impede but rather enhances oil synthesis. [15,16]. Additional research has demonstrated that the curtailment of starch synthesis can impede or diminish the accumulation of oil in seeds. Notably, the PGM enzyme, which plays a pivotal role in starch anabolism by facilitating the reversible conversion between G-1-P and G-6-P, has been shown to significantly impact oil content in *Arabidopsis* mutants with PGM gene knockout, resulting in a 40% reduction compared to wild-type seeds [17]. Anyway, the disruption of the AGP gene, responsible for catalyzing the initial stage of starch biosynthesis, resulted in a significant decrease in starch levels and a postponement of oil accumulation in Brassica napus, commonly known as rapeseed [18,19].

In this study, the fruits of the woody oil plant *S. paniculate* were selected as research material to further explore key genes in carbohydrate and lipid metabolism pathways. Through identification and annotation of functional unigenes related to these pathways, we have reconstructed their regulatory pathways based on information from key enzymes. Finally, the intricate molecular regulatory mechanism governing sugar metabolism has been elucidated, shedding light on its impact on oil biosynthesis in both seed and non-seed tissue of *S. paniculata* fruit.

## 2. Results

### 2.1. Temporal Pattern of Metabolite Contents

The morphological evolution of the *S. paniculata* fruit during its developmental period is illustrated in Figure 1A. At 10 days after flowering (DAF), the nascent fruitlet emerges, and from 10 DAF to 70 DAF, the fruit undergoes a rapid enlargement phase (REP) characterized by exponential growth. Subsequently, during the slow enlargement stage (SES) spanning from 70 DAF to 100 DAF, minimal changes in size were observed. And from 100 DAF to 120 DAF, the fruit hue shifted predominantly from a dull brown to a resplendent blue (fruit color transformation stage CTS). During the final phase (120 DAF to 140 DAF), the fruit’s epicarp began to contract as it reached maturity (the mature stage, MAS).

The alterations in sugar and oil contents within the pulp and seeds of *S. paniculata* at various stages are depicted in Figure 1B. Both fruit tissues exhibited a continuous accumulation of lipids throughout their development, albeit with differences in the rate of oil buildup during specific growth phases. The period of rapid oil accumulation in the pulp corresponds to the MAS stage (120 DAF to 140 DAF). And the sucrose content exhibited a decline during the REP and SES phases, spanning from 40 to 100 DAF. However, it experienced an abrupt surge in the CTS and MAS stages (100 to 140 DAF). In contrast, starch content displayed a gradual increase throughout the REP and SES periods before plummeting rapidly in the CTS and MAS phase.

Compared to pulp, the rapid accumulation of oil in seeds occurs during the SES (70 DAF to 100 DAF) and CTS (100 DAF to 120 DAF) stages. During seed development, the accumulation of sucrose and oil within the seeds increases from the REP to CTS stage (40 to 120 DAF), while starch content remains constant. In the MAS stage (120 DAF to 140 DAF), both sucrose and starch exhibit a gradual upward trend.

The fruit oil is primarily composed of saturated fatty acids, such as palmitic acid (C16:0) and stearic acid (C18:0), as well as unsaturated fatty acids, including oleic acid (C18:1), linoleic acid (C18:2), and linolenic acid (C18:3). The relative contents of the fatty acids differ significantly between the pulp and seed. In the pulp, the main FA components are C18:1 and C16:0, with a relative content increase from 19% to 52% for C18:1 and from 11% to 39% for C16:0 with fruit maturity. In contrast, *S. paniculata* seeds were found to contain mainly C18:1 and C18:2 fatty acids, with notable changes including a decrease in C18:2 content from 41% to 28%, and an increase in the percentage of C18:1 from 44% to 58%. Additionally, the relative content of C16:0 in seeds was observed to be approximately 30% lower than that present in pulp (Figure 2A).

For the total content change in unsaturated fatty acids (UFA) and saturated fatty acids (SFA), there was a significant variation in pulp SFA during development, ranging from 62% to 38%. However, few changes were observed in the UFA composition of total lipids in developing seeds, which remained stable at 88% of the total FA (Figure 2B). Additionally, medium-chain fatty acids (MCFA) such as C6:0, C8:0, C10:0, C11:0, and C12:0 were abundant in early stages of pulp development, but fewer MCFA were found in seed oil (Figure 2C).

The comprehensive analysis revealed that there was no significant alteration in oil and fatty acid composition during the fruit’s developmental stage from 40 to 90 DAF. However, a rapid accumulation of these components occurred from 90DAF-120DAF, accompanied by a change in fruit color and maximum pulp oil content. During this stage, there was a significant and rapid change in sugar content (either an increase or a decrease), indicating full maturity of the fruit at 140 days after flowering. Therefore, we selected 90, 120, and 140 days after flowering as representative periods of fruit development with notable differences in oil synthesis and sugar content, for further transcriptome analysis.

### 2.2. Functional Annotation of Sugar Metabolism and Lipid Synthesis Genes in S. paniculata Fruit

In this study, Illumina sequencing of cDNA libraries was performed on 18 samples representing different developmental stages of *S. paniculata* fruits, resulting in the acquisition of 51,002,774 raw reads (Appendix A). After filtering out impurities, 50,689,551 high-quality clean reads (99.39% of the raw reads) were obtained, on average. The content of Q30 and GC was 91.69% and 46.47%, respectively. The obtained clean read fragments were assembled using Trinity software, resulting in 124,923 non-redundant unigenes with an average length of 915 bp and an N50 of 1480 bp (Appendix A). The length distribution of unigene sequences ranged mainly from 200 to 3000 nt. As the sequence length increased, there was a gradual decrease in the number of unigenes without any significant disjunction observed (Appendix A). These findings suggest that high-quality RNA sequencing with good continuity was performed.

### 2.3. Functional Annotation of Non-Redundant Unigenes

To annotate the protein functions of unigenes, 124,923 sequences from all *S. paniculata* fruit samples were compared to protein databases (Nr, SwissProt, KEGG, and KOG) using BLASTX with an e-value threshold of <0.00001 to identify the protein with the highest sequence similarity. There were 64,445 Leukotan Unigenes annotated, accounting for 51.59% of the total number of unigenes, while the remaining 48.41% remained unannotated. This suggests that there is still more genetic information to be uncovered regarding the whole-fruit oil-producing woody plant *S. paniculate*. Among the hit unigenes, there were 56,535 (45.26%), 39,085 (31.29%), 59,963 (48.00%), and 47,535 (38.05%) that could be compared to KEGG, KOG, Nr, and Swissprot databases, respectively (Table 1).

BLAST matches were utilized to conduct a similarity analysis between *S. paniculata* unigenes and NR protein databases. The putative proteins accounted for 25.16%, 20.93%, 20.67%, and 18.27% in the four categories of low E-value, respectively (Appendix A). With reference to known proteins in the NR protein database, the BLAST match analyses were ensured to be accurate and reliable, as evidenced by the significant matches between *S. paniculata* unigenes and homology genes from *Quercus suber* (13,594 unigenes), *Camellia sinensis* (12,640 unigenes), *Actinidia chinensis* (4363 unigenes), and other plants (Appendix A).

In the meantime, COG (Clusters of Orthologous Groups of proteins) analysis was employed to ascertain the functions of the anticipated unigenes [20]. The 39,085 unigenes were classified into 25 functional categories (Appendix A). The most extensive cluster is “general function prediction only” (8305), which suggests that a considerable number of unknown genes in *S. paniculate* deposited in public databases have tremendous potential for exploration. The second largest cluster is “posttranslational modification, protein turnover, chaperones” (4877 unigenes), followed by “signal transduction mechanisms” (4381 unigenes) and “translation, ribosomal structure, and biogenesis” (3614 unigenes). However, the group of “cell motility” only contains 46 unigenes. Among all these unigenes, the group of “carbohydrate transport and metabolism” accounts for 2333 unigenes while the group of “lipid transport and metabolism” has 2054 unigenes.

In the KEGG database comparison results, 56,535 (50.97%) unigenes were annotated to 144 pathways across five primary categories and nineteen secondary categories: Metabolism, Genetic Information Processing, Environmental Information Processing, Cellular Processes, and Organismal Systems. Among them, the most highly annotated metabolic pathways include Carbohydrate Metabolism, Amino Acid Metabolism, Energy Metabolism, and Lipid Metabolism with corresponding unigene numbers of 3269, 2061, 1517, and 1566, respectively (Appendix A).

### 2.4. Functional Annotation of Sugar Metabolism and Oil Synthesis Genes in Symplocos paniculate Fruits

The volcano plot depicting differential gene expression between the pulp and seeds of *S. paniculata* is presented in Figure 3A. The identified differentially expressed genes were subjected to statistical analysis, revealing a total of 7943 genes exhibiting significant differences in expression levels during fruit development at 90 DAF, with 3396 up-regulated and 4547 down-regulated. In 120 DAF, a total of 10,763 differentially expressed genes were identified, with 4480 up-regulated and 6283 down-regulated. Similarly, in 140 DAF, there were a total of 24,363 differentially expressed genes detected, with the majority being up-regulated (18,673) and the remainder down-regulated (5690) (Figure 3B). Furthermore, as depicted in the Venn diagram presented in Figure 3C, of all the up-regulated unigenes, 1789, 1177, and 15,744 were found to be independently altered across the three group comparisons of pulp and seeds at 90, 120, and 140 DAF, respectively. In contrast, the expression levels of 1430, 2654, and 2862 unigenes were found to be down-regulated at 90, 120, and 140 DAF respectively.

The KEGG comparison of differential genes in the pulp and seeds of *S. paniculata* at different developmental stages (Figure 4) revealed that 335,501, and 1136 genes were found to be differentially expressed in carbohydrate metabolism across the developmental stages of 90, 120, and 140 DAF, respectively, for both pulp and seeds. In contrast, lipid metabolism exhibited differential expression in only a subset of genes with counts of 145, 231, and 594 across the same developmental stages, respectively. Further statistical analysis of differentially expressed genes involved in carbohydrate and lipid metabolism was performed (the results are shown in Appendix A), and the numbers of differentially expressed genes in the Glycolysis/Gluconeogenesis pathways, starch and sucrose metabolism, Citrate cycle, fatty acid metabolism pathway, and glycerophospholipid metabolism pathway were revealed.

### 2.5. Analysis of the Expression Pattern of Differentially Expressed Genes in the Pulp and Seeds of Symplocos paniculata

Through annotation of the KEGG metabolic pathway, we identified enzyme genes related to key pathways in sugar metabolism and oil biosynthesis. We then constructed differential expression profiles for key genes involved in carbohydrate metabolism and oil synthesis in *S. paniculata* (Figure 5), the metabolic pathways involved in sugar metabolism include Glycolysis/Gluconeogenesis, starch and sucrose metabolism, and the TCA cycle; while those involved in lipid metabolism encompass fatty acid biosynthesis, fatty acid elongation, and TAG assembly.

In 90 DAF: Among the key enzyme genes involved in starch and sucrose metabolism, the up-regulated enzyme genes Sucrose-Phosphate Synthase (SPS) and Sucrose Synthase (SUS) were associated with higher sucrose content in the seeds, while the slightly down-regulated enzyme genes Granule-Bound Starch Synthase (WAXY), Starch Branching Enzyme (GBE1), and Maltose Transglycosylase (malQ) were associated with lower pulp content. The majority of the remaining enzyme genes within the Glycolysis/Gluconeogenesis pathways and TCA cycle remained largely unaltered, with only a select few (PGK, PDH, MDH, etc.) experiencing down-regulation. This ultimately resulted in a slightly elevated rate of oil accumulation within the pulp. It is widely acknowledged that 3-oxoacyl-ACP reductase (KAR) and Glycerol-3-Phosphate O-Acyltransferase (GPAT) are pivotal enzyme genes in the pathways of oil synthesis related to lipid synthesis, whereas Fatty Acyl-ACP Thioesterase B (FATB) is intimately associated with the biosynthesis of medium-carbon chain fatty acids. The enzyme genes KAR and GPAT, among others, which were down-regulated, exhibited a close association with the rate of oil accumulation that was slightly higher in the pulps. Furthermore, FATB down-regulation resulted in a high content of medium-carbon chain fatty acids in the pulp, leading to 40% of medium-carbon chain fatty acids at 90 DAF.

In 120 DAF: The gene-encoding SUS and SPS enzymes, which are key players in starch and sucrose metabolism, were significantly up-regulated. Conversely, other enzyme genes such as malQ, SS, WAXY, and GBE1 were down-regulated. The expression of amylolytic enzyme gene malQ was particularly pronounced in the pulp tissue leading to a more rapid breakdown of starch therein, thereby promoting oil synthesis. In the pathways of Glycolysis/Gluconeogenesis and TCA cycle, the key enzyme genes including PGK, PK, PDH, MDH, and others were predominantly down-regulated. This suggests that the expression of related genes was more active in the pulp of *S. paniculate*. The expression of enzyme genes involved in the lipid synthesis pathway was predominantly down-regulated, including ACC, KASIII, KAR, HAD, EAR, and PAP. This suggests that these enzyme genes were highly expressed in the pulps of *S. paniculate* at 120 DAF, leading to a significant increase in oil accumulation rate from 59% to 165% at the beginning of 120 DAF.

In 140 DAF: The majority of enzyme genes, including PGK, PK, PDH, MDH, and others, were found to be up-regulated in the starch and sucrose metabolism pathways as well as the Glycolysis/Gluconeogenesis pathway and TCA cycle of *S. paniculate*. Interestingly, related enzyme gene expressions in the pulp of *S. paniculate* were observed to be lower than those in seeds, leading to a more rapid accumulation of oil in seeds. In the pathways associated with lipid synthesis, the majority of enzyme genes in both the pulp and seeds of *S. paniculate* were up-regulated, including KAR, GPAT, LPAAT, PAP, and DGAT. During this stage, there was a shift in expression patterns for key enzyme genes in both pulp and seeds which resulted in a significant increase in oil accumulation within the seeds.

### 2.6. Real-Time Fluorescence Quantitative PCR Validation

To further validate the precision of transcriptome sequencing outcomes for *S. paniculate*’s pulp and seed, we performed real-time PCR on four differential genes involved in sugar metabolism (PGK3, PKP1, PDH-E1, MDH) and four differential genes related to lipid metabolism (FATB, DGAT3, MAT, LPAAT), which were identified from RNA-Seq data. The gene expression levels were assessed by qRT-PCR and found to be consistent with the RNA-seq results (Figure 6). The relative expression levels of eight genes displayed a significant correlation with the RNA-seq analysis results (Appendix A), further validating the reliability of the *S. paniculate* pulp transcriptome sequencing expression profile and demonstrating that RNA-seq technology is a viable approach for differential gene expression pattern analysis of both *S. paniculate* pulp and seed transcriptome sequencing using DESeq technology.

## 3. Discussion

### 3.1. The Correlation between the Sugars and Oil Content during Fruit Development

Carbohydrate substances form the basis for oil accumulation [21], and although there is a certain correlation between sugar and oil accumulation, the intricate interplay between these two processes cannot be reduced to mere changes in content. Real-time detection of sugar and oil content in *S. paniculate* fruit has revealed stage differences in the accumulation patterns of sucrose, starch, and oil within both pulp and seeds. The development of other oil seeds is closely associated with the accumulation of sugar at various stages [7]. Although there is not a particularly strong correlation between oil content, starch content, and sucrose content in the early developmental stage of *S. paniculate* fruit, this may be attributed to the fact that sucrose serves as both a precursor for starch [22] and an independent storage compound for energy [23]. In the slow expanding stage (SES), there was a negative correlation between oil and starch content in both pulp and seeds, which is consistent with the relationship observed during the growth and development of *Xanthoceras sorbifolia* fruit seeds where soluble sugars mainly transform oil and starch [7]. During the fruit color transformation stage (CTS), there exists a positive correlation between oil content and starch content in both pulp and seeds, whereby the synthesis of starch will facilitate the synthesis of oil during this period [16]. The gene expression profile of Jatropha curcas seed development highlights the pivotal role of starch metabolism and other pathways in regulating oil synthesis [24]. Additionally, studies have shown that an accumulation of sugars during the middle stage of fruit development facilitates the production of oils and other substances in later stages [25]. During fruit development, there is a negative correlation between sugar and fat content [26]. Previous studies on Olea europaea and Ricinus communis have demonstrated that sucrose inhibits oil synthesis [27]. The breakdown of sucrose facilitates oil production. The synthesis of sucrose, starch, and oil appears to be developmentally regulated with no definitive transformational relationship among them.

### 3.2. Key Genes Involved in the Sugar Metabolism Pathway Play a Crucial Role in Regulating Lipid Synthesis

In terms of the sugar metabolism pathway, both Sucrose-Phosphate Synthase (SPS) and Sucrose Synthase (SUS) play pivotal roles in both the synthesis and catabolism of sucrose [28]. The expression of the SPS gene was significantly up-regulated in the seeds of *S. paniculate* at 90 DAF and 120 DAF, as compared to that observed in the pulp. In a study on SPS gene families involved in the regulation of sucrose accumulation in sugarcane (Saccharum officinarum), it was observed that up-regulation of SPS expression can enhance enzyme activity, thereby facilitating sucrose synthesis [29]. And in another study during the investigation of photosynthate accumulation, distribution, and transport mechanisms in *Camellia oleifera*, it was observed that SPS and SUS genes associated with carbohydrate synthesis were up-regulated during periods of rapid oil synthesis [30]. Regarding the pathway of starch synthesis, the Granule-Bound Starch Synthase (WAXY), and Starch Branching Enzyme (GBE1) genes are essential, with WAXY primarily regulating amylose biosynthesis and GBE1 playing a crucial role in amylopectin production, both contributing to overall starch synthesis. Conversely, Maltose Transglycosylase (malQ) operates on starch degradation. The starch content in the pulp was found to be lower than that of the seeds. However, at 90 and 120 days after flowering, genes related to starch metabolism were expressed more strongly in the pulp compared to those in the seeds. Additionally, increased expression of malQ within the pulp facilitated oil synthesis and led to a remarkable increase of 165.51% in oil accumulation between 120 and 130 days after flowering. At 140 DAF, the expression of malQ was found to be more pronounced in the seeds, leading to a greater reduction in starch content. A similar phenomenon was observed in the Xanthoceras sorbifolia fruit [12].

In this study, the Phosphoglycerate Kinase 3 (PGK3), pyruvate kinase (PK), PDH-E1, and Malate Dehydrogenase (MDH) were genes that exhibited similar expression in the glycolysis and tricarboxylic acid pathways between pulp and seed. The up-regulation of numerous enzymes involved in glycolysis, such as PGK and PK, may facilitate the production of ample pyruvate necessary for high oil synthesis in oilseed plants [31]. A highly expressed plastid-localized NAD-MDH was found to be significantly correlated with fatty acid synthesis during the fruit development of Lindera glauca [32], indicating its crucial role in energy homeostasis for fruit fatty acid synthesis. This is supported by mutant studies of plastidial NAD-MDH in developing *Arabidopsis thaliana* seeds [33,34]. The Pyruvate Dehydrogenase (PDH) enzyme, which plays a crucial role in glucose metabolism and lipid synthesis, facilitates the conversion of pyruvate into acetyl-CoA for fatty acid synthesis. This study found a positive correlation between PDH activity and oil content in both the pulp and seeds of *S. paniculate*. It has demonstrated a significant increase in the expression level of PDH during seed oil accumulation in *Arabidopsis thaliana*. Inhibition of negative regulators of PDH activity, such as pyruvate dehydrogenase kinase (PDK), promotes the conversion of pyruvate to acetyl-CoA and inhibits PDH activity, thereby increasing carbon source utilization for oil synthesis pathway [35].

### 3.3. The Pivotal Genes in the Lipid Metabolism Pathway Facilitate the Biosynthesis of Oil from Symplocos paniculate Fruit

In the pathway of lipid synthesis, significant disparities were observed between pulp and seed in terms of FATB, HAD, KAR, DGAT, and other genes. Acyl ACP thiolipase B (FATB) and 3-keto-acyl ACP synthetase III (KAS III) are closely linked to fatty acid chain elongation. Deletion of KASIII and FATB in *Brassica napus* resulted in a significant augmentation in the medium-chain fatty acids proportion within the offspring double mutants [8]. In the differential gene expression between pulp and seed at 90DAF, FATB was found to be significantly down-regulated in the seed, while abundantly expressed in the pulp. In previous research, FATB has been shown to facilitate the synthesis of medium-chain fatty acids such as C8 and C12 [36], leading to their abundant production in the pulp at this stage. This aligns with FATB’s role in fatty acid biosynthesis in Cocos nucifera [37]. Meanwhile, HAD and KAR are pivotal players in fatty acid synthesis, with overexpression of the KAR gene significantly boosting oil content in cottonseed [38]. At 120 DAF, the expression of KAR in pulp is significantly elevated compared to that in seed, and this heightened expression of KAR directly correlates with an increase in oil growth rate at 120 DAF. However, by 140 DAF, the expression of KAR in seeds surpasses that found within pulp tissue, resulting in a greater concentration of oil content within the seeds as opposed to the pulp. Glycerol 3-phosphate acyltransferase (GPAT), lysophosphatidyl phosphatidyl transferase (LPAAT), phosphatidic acid phosphatase (PAP), and glycerol diester acyltransferase (DGAT) are the pivotal enzymes that impact the efficiency of triglyceride assembly. The introduction of a specific DGAT into soybean leads to a significant increase in oil content [39]. At 120 DAF, the majority of genes involved in fatty acid synthesis exhibited high expression levels within the pulp. However, solely the TAG assembly-related enzyme gene PAP was expressed more prominently within the pulp than in the seed, ultimately contributing to an increase in oil growth rate from 120 DAF onwards. By contrast, at 140 DAF, oil content within the seed surpassed that found within the pulp—a phenomenon potentially linked to heightened KAR and assembly-related gene expression levels present specifically within said pulp.

## 4. Materials and Methods

### 4.1. Plant Material

Fresh fruits of *S. paniculata* were harvested every 10 days after flowering (DAF) in 2021 from Hunan Academy of Forestry (28°06′55″ N; 113°03′16″ E), Changsha city, Hunan Province, China. The growth conditions, selection criteria, and sampling method for the plants followed those previously described [40]. For each phase, the pulp and seeds were manually separated using tweezers. However, the inner seed coat from 10 to 30 DAF fruit was not yet mechanized, making it challenging to completely separate the pulp and seeds. Therefore, metabolite content analysis was initiated from 40 DAF when the seed coat began to mechanize, allowing for easy separation of the pericarp and seeds. And all fruit tissue (separated pulp and seed) was divided into quarters for transcriptome sequencing, oil extraction, physiochemical properties analysis, and qRT-PCR verification. Therefore, a quarter of the samples were flash-frozen under liquid nitrogen after peeling into seeds and peels and wrapping in tinfoil and then stored at −80 °C for later analysis, and a quarter of the samples were stored at −80 °C for assays of physicochemical properties, while the others were divided into peels and seeds and dried in an oven at 60 °C for 4 days until their weight did not change, and ground in an hammer mill (RT-02 miniature plant pulverizer) and sieved through serial stainless steel sieves to 0.15 ram.

### 4.2. Sucrose and Starch Contents Measurement

The contents of sucrose and starch in the samples were determined using anthrone colorimetry [36]. And each measurement was conducted with three biological replicates. The physiological characteristics were statistically evaluated through one-way analysis of variance (ANOVA) with Duncan’s multiple comparison test (*p* < 0.05) in IBM SPSS Statistics 27.

### 4.3. Oil Content Determination

Four grams each of dried and ground peels and seeds were mixed with 60 mL of petroleum ether (99.7%, boiling point range 30 to 60 °C) in an SZE-101 Fat Analyzer (Shanghai Shine Jan Instruments Co. Ltd., Shanghai, China). The mixture was heated to reflux at 65 °C for six hours with vigorous stirring, then filtered and evaporated to dryness using a rotary vacuum evaporator at 62.5 °C, yielding the oil. After the extraction process, the samples were subjected to a drying procedure in an oven set at 110 °C for four hours until their weight stabilized [41]. The final weight was recorded and used to calculate the oil content (w) using the following formula:(1)$$x=\frac{\left(Ma-Mb\right)}{Ma}\;\times \;100\%$$
where *Ma* represents the weight of the sample prior to extraction and *Mb* denotes the weight of the sample subsequent to extraction (with a precision determined at 0.0001 g). Each test was conducted with three biological replicates.

### 4.4. Fatty Acid Composition Determination

About 0.06 g of the oil sample was weighed and mixed with 4 mL of isooctane, followed by the addition of 200 mL of a potassium hydroxide methanol solution (2 mol/L). The mixture was vigorously shaken for 30 s before adding 1 g of anhydrous NaHSO4. After precipitation and filtration, the supernatant was left for measurement.

The oil composition was analyzed using a Shimadzu 2030 gas chromatograph equipped with a quartz glass capillary column (0.25 mm × 100 m, film thickness of 0.25 μm). Helium was used as the carrier gas at a flow rate of 1.1 mL/min and split ratio of 1:100. The initial oven temperature was optimized at 100 °C for 13 min and ramped up to 180 °C at a rate of 10 °C/min, followed by an increase to 200 °C at a rate of 1 °C/min. The injector temperature was maintained at a constant level of 300 °C. Finally, the temperature was raised to 230 °C with a ramp rate of 4 °C/min and held for another period lasting for approximately ten and half minutes. The GC fatty acid profile was determined using Supelco 37 Component FAME Mix as the standard sample, and the results were calculated by normalizing peak areas. The relative content of each fatty acid was then calculated by normalizing its peak area to the total chromatogram [42].

### 4.5. Transcriptome Sequencing and Analysis

The RNA of both pulp and seed from *S. paniculate* fruit was extracted using a DNA extraction kit (Sangon Biotech., Shanghai, China). Both the purity and concentration of the total RNA were assessed with a Nanodrop 2000 ultramicroscopy spectrophotometer. Subsequently, the cDNA library was prepared in accordance with the mRNA-Seq sample preparation kit (Illumina Biotechnology Company, San Diego, CA, USA), followed by sequencing on an Illmina2500 (HiSeq 2500, Illumina) platform [43].

The sequencing work was commissioned by Guangzhou Gidio Biotechnology Co., Ltd., Guangzhou, China. Quality control was performed using fastp [44], and high-quality data were obtained after filtering low-quality data. Reads were assembled using Trinity software [43], and the integrity of the assembled data was evaluated and confirmed with BUSCO [45]. Functional annotation was performed by matching high-quality bases to the KEGG public database, while quantification analysis utilized RSEM [46].

The input data for gene differential expression analysis consisted of read count data obtained from gene expression level analysis. DESeq2 [47] software was utilized, and after multiple verifications, the screening difference factor was determined to be greater than 2. The key enzyme genes involved in sugar accumulation metabolism and oil synthesis in both pulp and seeds of *S. paniculate* were compared using FPKM value and expression analysis with an FDR value less than 0.05, followed by differential expression analysis between the pulp and seed tissues.

### 4.6. Validation of RNA-seq by Quantitative PCR

Eight key genes related to carbohydrate metabolism and lipid biosynthesis pathways (Unigene0054085, Unigene0010619, Unigene0054085, Unigene0054085, Unigene0054085, Unigene0054085, and Unigene0054085) were selected as target genes, respectively (Appendix A), with primers designed using Primer Premier 5.0 (Premier Biosoft International, Palo Alto, CA, USA). ALB (albumin) and ETIF3H (Eukaryotic translation initiation factor 3) served as internal reference genes for quantitation. The reaction conditions for all samples included an initial denaturation step at 95 °C for 2 min, followed by 40 cycles of amplification consisting of a denaturation step at 95 °C for 15 s, annealing and extension steps at 60 °C for 15 s, and a final denaturation step at 95 °C for 15 s. and each reaction was repeated three times. The relative expression levels of each unigene were determined using the comparative cycle threshold (ΔΔCt) method.

## 5. Conclusions

In this study, we conducted a quantitative analysis of sugar and oil in *S. paniculate* fruits at different developmental stages, as well as a transcriptome analysis to investigate key gene regulation. A total of 124,923 unigenes were obtained from transcriptome sequencing of *S. paniculate* pulp and seed. Among them, 51.59% were successfully annotated to public databases. Based on KEGG database annotation, we identified 335,501, and 1136 differentially expressed genes involved in sugar metabolism at 90 DAF, 120 DAF, and 140 DAF, respectively. Additionally, we also screened out 145,231, and 594 differentially expressed genes related to oil metabolism pathway, respectively.

Preliminary screening has identified key regulatory genes involved in sugar metabolism and oil synthesis in both the pulp and seed of *S. paniculate*. Specifically, PGK3, PKP1, PDH-E1, MDH, and malQ were found to be predominantly expressed during the slow expanding stage and over-mature stage in the pulp. Conversely, genes related to sucrose catabolism such as SUS were mainly expressed in seeds after reaching maturity. The oil synthesis rate of pulp during the slow expansion and over-maturity stages is primarily influenced by the KAR, HAD, and TAG assembly enzyme gene PAP related to oil synthesis. At 140 DAF, LPAAT plays a significant role in this process. The high expression of FATB in pulp leads to an abundance of carbon chain fatty acids while GPAT critically regulates lipid synthesis rates. Based on the findings of this investigation, a comprehensive understanding of lipid synthesis has been achieved through an analysis of sugar content and metabolism. The interplay between sugar metabolism and lipid biosynthesis in various regions was explored, while key regulatory genes were scrutinized to provide a solid theoretical foundation for the rational and efficient development of *S. paniculate* resources.

## Figures and Tables

**Figure 1 plants-12-02703-f001:**
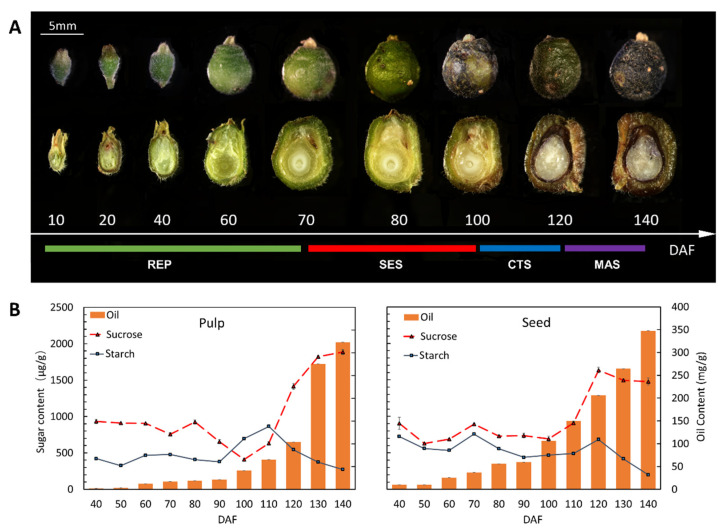
Change trend of sucrose content, starch content, and oil content in pulp and seed of *Symplocos Paniculata.* (**A**). External morphology of fruits at different developmental stages; (**B**). Developmental changes in metabolite content of pulp and seed. The oil content was quantified by solvent extraction, and sucrose and starch content were determined using anthrone colorimetry. Each test was conducted with three biological replicates.

**Figure 2 plants-12-02703-f002:**
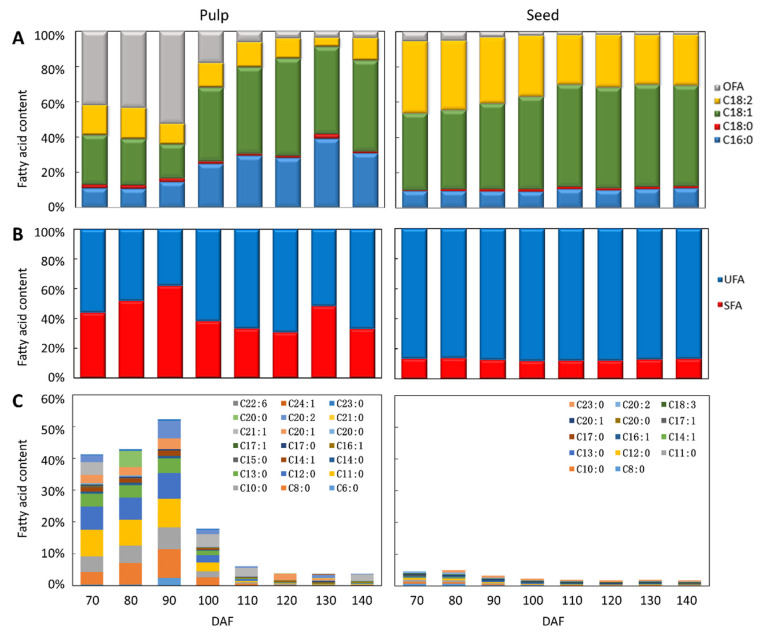
Change trend of fatty acid components in pulp and seed of *Symplocos Paniculata* during distinct developmental stages. (**A**). Revisions in the composition of primary fatty acids; (**B**). Revisions in the composition of UFA and SFA; (**C**). Revisions in the composition of Other Fatty Acids (OFA), OFA is except major FA, and the % of each OFA < 1% of total FA in mature fruit oil. The fatty acid composition of pulp and seed oils was determined using gas chromatography-mass spectrometry (GC-MS) with three replicates.

**Figure 3 plants-12-02703-f003:**
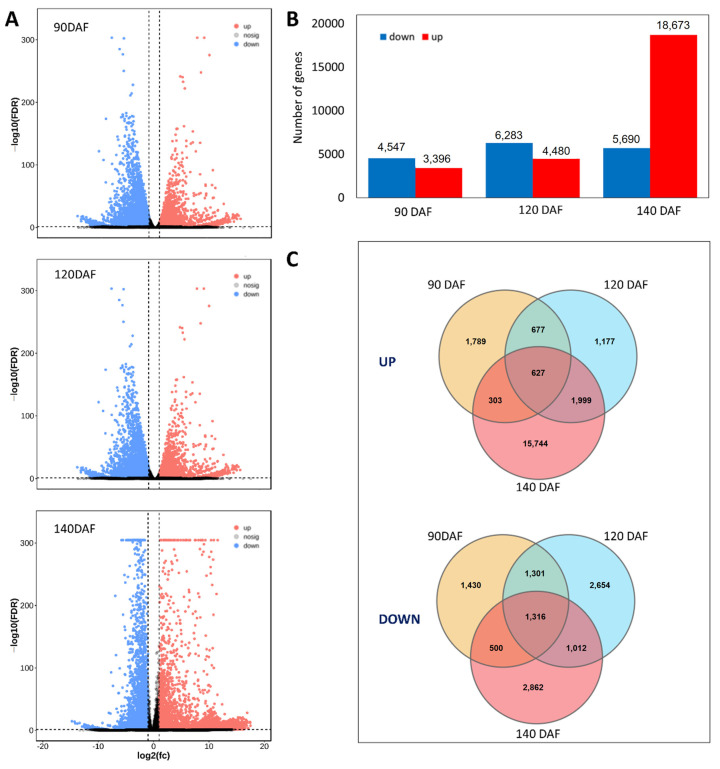
Differentially expressed transcripts between pulp and seed. (**A**). Volcano map of differentially expressed genes, (**B**). number of down-regulated genes, (**C**). Venn diagram of up and down regulated unigenes involved.

**Figure 4 plants-12-02703-f004:**
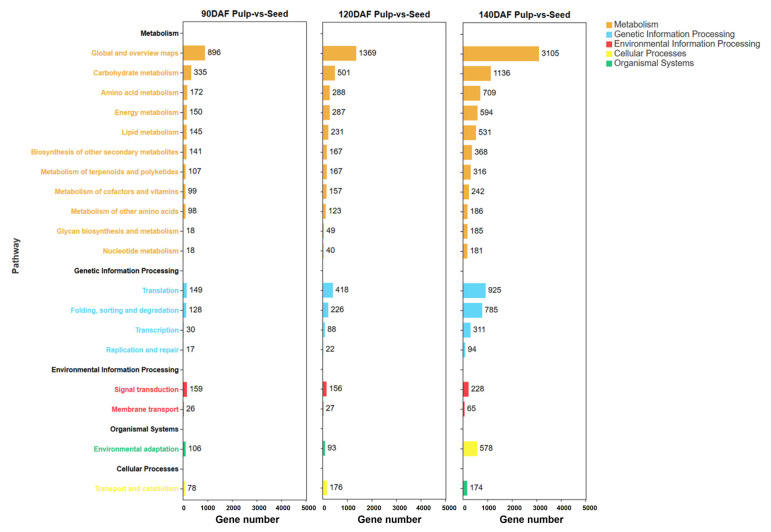
Classification of KEGG metabolic pathway in the pulp and seeds of *Symplocos paniculata* at different developmental stages.

**Figure 5 plants-12-02703-f005:**
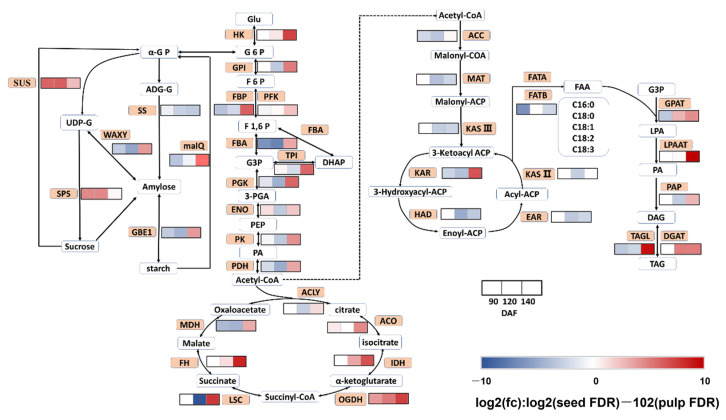
Key gene expression patterns in carbohydrate metabolism and lipid metabolism pathways in the pulp and seeds of *Symplocos paniculate.* SUS: Sucrose Synthase; UDP-G: uridine 5′-diphosphoglucose disodium; SPS: Sucrose-Phosphate Synthase; WAXY: Granule-Bound Starch Synthase; SS: Starch Synthase; GBE1: Starch Branching Enzyme; HK: hexokinase; GPI: glucose-6-phosphate isomerase; FBP: fructose-1,6-bisphosphatase; PFK: 6-phosphofructokinase 1; FBA: fructose-bisphosphate aldolase, class I; TPI: triosephosphate isomerase; PGK: phosphoglycerate kinase; ENO: enolase; PK: pyruvate kinase; PDH: pyruvate dehydrogenase; ACLY: ATP citrate (pro-S)-lyase; ACO: aconitate hydratase; IDH: isocitrate dehydrogenase (NADP+); OGDH: 2-oxoglutarate dehydrogenase(E1); LSC: succinyl-CoA synthetase alpha subunit; FH: fumarate hydratase; MDH: malate dehydrogenase; ACC: acetyl-CoA carboxylase; MAT: malate dehydrogenase; KAS III:3-oxoacyl-[ACP] synthase III; KAS II:3-oxoacyl-[ACP] synthase II; EAR: enoyl-[ACP] reductase I; HAD: 3-hydroxyacyl-[ACP] dehydratase; KAR: 3-oxoacyl-[ACP] reductase; FATA: fatty acyl-ACP thioesterase A; FATB: fatty acyl-ACP thioesterase B; GPAT: glycerol-3-phosphate O-acyltransferase; LPAAT: lysophospholipid acyltransferase; PAP: phosphatidate phosphatase; TAGL: triacylglycerol lipase; DGAT: diacylglycerol O-acyltransferase.

**Figure 6 plants-12-02703-f006:**
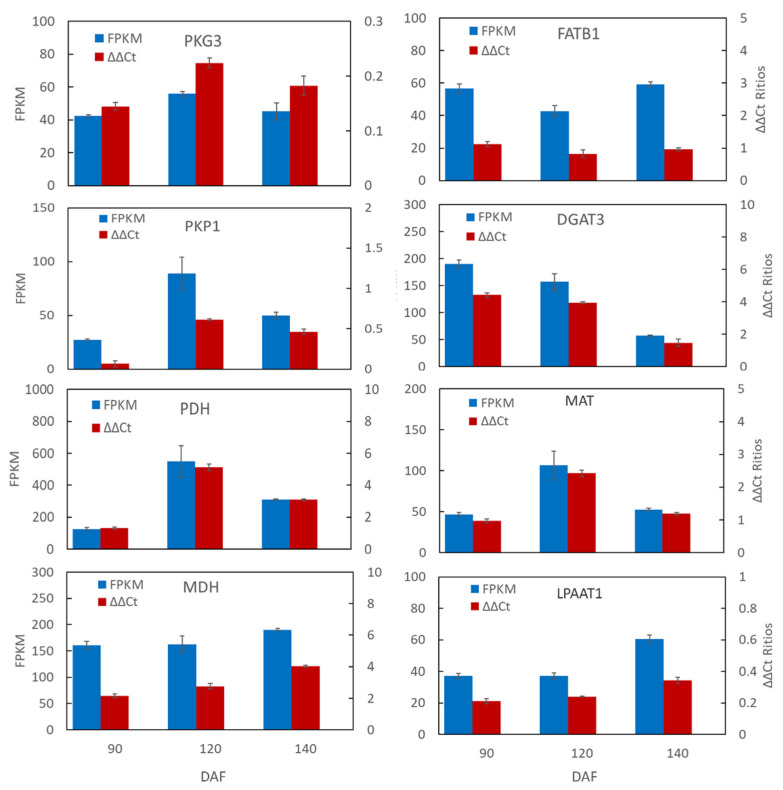
Validation of eight candidate genes related to sugar and lipid metabolism in *Symplocos paniculata* using quantitative RT-PCR. Results represent the mean (±SE) of three biological replicates. Error bars represent the standard error of three biological replicates.

**Table 1 plants-12-02703-t001:** Functional annotation of *Symplocos paniculata* unigenes in public protein databases.

Annotated Databases	Number of Unigenes	Percentage of Annotationed Unigenes (%)
Annotated in Nr	59,963	48.00
Annotated in KEGG	56,535	45.26
Annotated in KOG	39,085	31.29
Annotated in Swissprot	47,535	38.05

## Data Availability

The data that support the findings of this study are available from the corresponding author upon reasonable request.

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
