# Peer review of "Transcriptome Analysis Unveiled the Intricate Interplay between Sugar Metabolism and Lipid Biosynthesis in Symplocos paniculate Fruit"

_plants, 2023, doi:10.3390/plants12142703_

Round 1

Reviewer 1 Report

The authors did an oil, starch, sucrose analysis of the seed and pulp of Symplocos paniculata.
They also annotated the transcriptome and made a transcriptome analysis of different developmental stages of S. paniculata. They try to analysis key elements of sugar and lipid analysis.
It is a nice paper

There are some issues of editing. Figure captions are too short. One sentence about experimental procedure is needed. This is true for all figures.
The authors write “note” in the figure caption which is not requested.
Abstract
14. delete « novel »Ò
16. Delete « additionally »
Sentence line 20/21. Very vague
End of abstract : what is the conclusion of this study?

INTRO

81/ here again a conclusion on what you discovered is lacking. What is the main result of your study?

RESULTS
Figure 1 caption. One sentence about the way you quantified oil, starch and sucrose is needed.
Lines 109, 131 . delete “note”. Description of A and B goes with the figure caption.
Figure 2 caption. One sentence about the way you quantified FA is needed. What about repetitions
Figure 2 description: what about  descriptions of OFA?
Table 1/ % of what?  Here again caption is not informative enough.
Figure 4. write E-value and not evalue
Line 170. What is “COG”
Figure 5. cannot the categories be sorted from most abundant to less abundant? What is the current logic of the sorting?

494: catabolism is better than decomposition

Author Response

Response to Reviewer 1 Comments

Abstract

Point 1: Abstract,line 14. delete « novel »

Response 1: The “novel” was deleted. See line 14.

Point 2: Line 16. Delete « additionally »

Response 2: The “additionally” was deleted. See line 16.

Point 3: Sentence line 20/21. Very vague

Response 3: The abstract was rewrite,so the sentence of line 20/21 was deleted in abstract.

Point 4: End of abstract : what is the conclusion of this study?

Response 4: The conclusion was added in abstract. See line 20-28.

Introduction

Point 5: Line 81/ here again a conclusion on what you discovered is lacking. What is the main result of your study?

Response 5: The conclusion of this study was added,see line 87-89.

Results

Point 6: Figure 1 caption. One sentence about the way you quantified oil, starch and sucrose is needed.

Response 6: The methods for the determination of oil and sugar content were added in Figure1 caption, see line 118-122.

Point 7: Lines 109, 131 . delete “note”. Description of A and B goes with the figure caption.

Response7: The “note” in all Figures caption were deleted. See each figure caption.

Point 8: Figure 2 caption. One sentence about the way you quantified FA is needed. What about repetitions.

Response 8: The method for the determination of fatty acids was added in Figure 2 caption, see line 151-154.

Point 9: Figure 2 description: what about descriptions of OFA?

Response 9: The descriptions of OFA was added,see Figure 2 description (line 151-152).

Point 10: Table 1/ % of what?  Here again caption is not informative enough.

Response 10: The “Percentage of Annotationed Unigenes” was added in Table 1.

Point 11: Figure 4. write E-value and not evalue

Response 11 :All “evalue” in Figure 4 was changed to E-value, see figure S2.

Point 12: Line 170. What is “COG”

Response 12: The full name of COG was added, see line 187.

Point 13: Figure 5. cannot the categories be sorted from most abundant to less abundant? What is the current logic of the sorting?

Response 13: The categorization has been revised to arrange items in descending order of abundance. See Figure S3.

Point 14: Line 494: catabolism is better than decomposition

Response 14: The term "decomposition" was substituted with "catabolism", see line 511.

Reviewer 2 Report

This research article entitled "Transcriptome Analysis Unveiled the Intricate Interplay between Sugar Metabolism and Lipid Biosynthesis in Symplocos paniculate Fruit" describes the identification of the critical DEGs in specific metabolic pathways (e.g., TCA, glycolysis, and starch) underlying the quantification of sugar/starch and the lipid (e.g., fatty acid) biosynthesis during the fruit (pulps and seeds) development of the oil tree. The results prompted crosstalk between lipid and sugar metabolic pathways that might potentially modulate the oil composition and accumulation, leading to a bioengineering relevance. The manuscript's writing style is good, containing much broad information of interest. However, several significant concerns are the primary focus of the research presented and the underlying magnitude. Some suggestions to improve the manuscript are given below.

Abstract:

The abstract should be an objective representation of the article based on the criteria with (1) Background: Place the question addressed in a broad context and highlight the purpose of the study; (2) Methods: briefly describe the main methods or treatments applied; (3) Results: summarize the article's main findings; (4) Conclusions: indicate the main conclusions or interpretations.

Line 24: what specific mechanisms were presented based on the results and conclusion if no functional analyses of specific gene families? The author should address the significance and the informed application of the conducted research and the related novel findings.

Introduction:

Line 47-48: "In general, the metabolism of carbohydrates is intimately linked to the biosynthesis of lipids." How is the detailed intimate link or molecular basis between carbohydrates and lipids biosynthesis (e.g., pathways, genes, intermediates, or metabolite transformation)? Several critical references should be provided to support the above notion.

Line 49-50: The sentence is confusing. Given the intimate link between carbohydrates and lipids, how is it possible that "the relationship between oil and carbohydrate remains elusive"?

Line 57-60 and Line 65-66: The citation is missing, such as typos and writing format (e.g., legend, abbreviation, plant scientific name, genes, and proteins) should be double-checked carefully!

Line 60-61: It is not very clear! The conclusion "it is insufficient to analyze the effect of sugars on lipid synthesis solely through correlation analysis of sugar and oil content changes" was deduced from the sucrose transformation between source (leaves) and sink (fruits); how it could be logically possible!

Results

Line 85-86: Based on the image in Fig 1A, the author should specify the technique for isolating pulp and seed from 10-60 DAF. The author should explain why three selected DAFs (90, 120, and 140) are used for the DEGs analyses and specify compared groups in pulp vs. seed or seed vs. pulp.

Please move Fig 3-6 to the supplement materials, as Table 1 has summarized the functional annotation using different databases.

Line 204-206: The sentence is confusing, considering revision.

Line 288-290: Why chose these genes for the qRT-PCR evaluation? Many DEGs were identified in carbohydrate and lipid metabolism based on Fig 9. Please explain.

Discussion

This part is impoverished, and some critical issues have not been argued significantly. The authors discussed sugar accumulation in correlation with lipid biosynthesis via the catalysis of enzymatic genes in the metabolic pathway. However, no further reports or functional experiments were integrated to support the influence between the enzyme genes and metabolites in carbohydrate and lipid biosynthesis pathways.

Materials and Methods:

Authors should make a significant effort to clarify the methodology (e.g., metabolites isolation and RNA extraction) and provide the statistical analyses, including the biological repeats with how many replicates in each experiment, particularly for detecting the compound and qRT-PCR.

Line 407: The citation is missing.

Line 476: typos and format

Some writing formats should be extensively focused on.

Author Response

Response to Reviewer 2 Comments

Abstract

Point 1: The abstract should be an objective representation of the article based on the criteria with (1) Background: Place the question addressed in a broad context and highlight the purpose of the study; (2) Methods: briefly describe the main methods or treatments applied; (3) Results: summarize the article's main findings; (4) Conclusions: indicate the main conclusions or interpretations.

Response 1: The abstract has been revised according to the aforementioned criteria. See line 14-31.

Point 2: Line 24: what specific mechanisms were presented based on the results and conclusion if no functional analyses of specific gene families? The author should address the significance and the informed application of the conducted research and the related novel findings.

Response 2: The study provided clarification on the specific mechanisms of sugar metabolism that impact oil synthesis, and highlighted its significance. See line 28-31.

Introduction

Point 3:Line 47-48: "In general, the metabolism of carbohydrates is intimately linked to the biosynthesis of lipids." How is the detailed intimate link or molecular basis between carbohydrates and lipids biosynthesis (e.g., pathways, genes, intermediates, or metabolite transformation)? Several critical references should be provided to support the above notion.

Response 3: More references of [6]-[10] were added to support the claim of “carbohydrates is linked to oil biosynthesis”, see line 55-62.

Point 4: Line 49-50: The sentence is confusing. Given the intimate link between carbohydrates and lipids, how is it possible that "the relationship between oil and carbohydrate remains elusive"?

Response 4: The sentence was modified, see line 63-64.

Point 5: Line 57-60 and Line 65-66: The citation is missing, such as typos and writing format (e.g., legend, abbreviation, plant scientific name, genes, and proteins) should be double-checked carefully!

Response 5: The sentence in lines 57-60 has been removed, while the citation [14] in lines 65-66 has been added (refer to lines 74-76). Additionally, all typographical errors and formatting issues have been corrected.

Point 6: Line 60-61: It is not very clear! The conclusion "it is insufficient to analyze the effect of sugars on lipid synthesis solely through correlation analysis of sugar and oil content changes" was deduced from the sucrose transformation between source (leaves) and sink (fruits); how it could be logically possible!

Response 6: This sentence of line 60-61 was modified, see line 66-70.

Results

Point 7: Line 85-86: Based on the image in Fig 1A, the author should specify the technique for isolating pulp and seed from 10-60 DAF. The author should explain why three selected DAFs (90, 120, and 140) are used for the DEGs analyses and specify compared groups in pulp vs. seed or seed vs. pulp.

Response7: (1) The technique for isolating pulp and seed from 10-60 DAF was added in  Materials and Methods part. See line 418-422. (2) The 90,120 and 140 DAF were selected for further transcriptome analysis due to the significant difference in metabolite changes between these three stages, which is suitable for differential expression analysis. Relevant explanations have been added. See line 140-147.

Point 8: Please move Fig 3-6 to the supplement materials, as Table 1 has summarized the functional annotation using different databases.

Response 8: The Fig 3-6 has been relocated to the Supplement Materials and its title has been revised to figures S1, S2, S3, and S4 respectively.

Point 9: Line 204-206: The sentence is confusing, considering revision.

Response 9: The sentence was revised, see line 217-220.

Point 10: Line 288-290: Why chose these genes for the qRT-PCR evaluation? Many DEGs were identified in carbohydrate and lipid metabolism based on Fig 9. Please explain.

Response 10: In this study, candidate genes were selected for qRT-PCR validation based on their pathway coverage and random expression pattern. Representative genes from the glycolysis pathway (PGK, PKP, PDH) were chosen, along with key genes involved in the tricarboxylic acid cycle (MDH), fatty acid synthesis (MAT, FATB), and TAG assembly (LPAAT), which were subsequently validated. Additionally, we took into account the FPKM expression patterns of randomly selected genes, which included those with a consistently decreasing trend (DGAT), consistently increasing trend (LPAAT and MDH), initially increasing then decreasing trend (PGK, PDH, MAT) and initially decreasing then increasing trend (FATB). As such, we consider our selection of qRT-PCR validation genes to be reasonable.

Discussion

Point 11: This part is impoverished, and some critical issues have not been argued significantly. The authors discussed sugar accumulation in correlation with lipid biosynthesis via the catalysis of enzymatic genes in the metabolic pathway. However, no further reports or functional experiments were integrated to support the influence between the enzyme genes and metabolites in carbohydrate and lipid biosynthesis pathways.

Response 11 :The discussion was revised to include additional details regarding the interplay between enzyme genes and metabolites in the biosynthesis pathways of carbohydrates and lipids. See line 317-365.

Point 12: Authors should make a significant effort to clarify the methodology (e.g., metabolites isolation and RNA extraction) and provide the statistical analyses, including the biological repeats with how many replicates in each experiment, particularly for detecting the compound and qRT-PCR.

Response 12: All the methodology was carefully clarified, and more detail about statistical analysis was added. See line 433,449, 495-496.

Point 13: Line 407: The citation is missing.

Response 13: The missing citation was added, see line 418.

Point 14: typos and format

Response 14: All typos and format were corrected. 

Comments on the Quality of English Language

Point 15: Some writing formats should be extensively focused on.

Response 15: All English language formats were carefully checked and revised.
